# CRISPR/Cas9 targeting events cause complex deletions and insertions at 17 sites in the mouse genome

Ha Youn Shin[1,2,*,**], Chaochen Wang[1,*], Hye Kyung Lee[1,3,*], Kyung Hyun Yoo[1,4], Xianke Zeng[1], Tyler Kuhns[1], Chul Min Yang[1], Teresa Mohr[1], Chengyu Liu[5] & Lothar Hennighausen[1,**]

Although CRISPR/Cas9 genome editing has provided numerous opportunities to interrogate the functional significance of any given genomic site, there is a paucity of data on the extent of molecular scars inflicted on the mouse genome. Here we interrogate the molecular consequences of CRISPR/Cas9-mediated deletions at 17 sites in four loci of the mouse genome. We sequence targeted sites in 632 founder mice and analyse 54 established lines. While the median deletion size using single sgRNAs is 9 bp, we also obtain large deletions of up to 600 bp. Furthermore, we show unreported asymmetric deletions and large insertions of middle repetitive sequences. Simultaneous targeting of distant loci results in the removal of the intervening sequences. Reliable deletion of juxtaposed sites is only achieved through two-step targeting. Our findings also demonstrate that an extended analysis of F1 genotypes is required to obtain conclusive information on the exact molecular consequences of targeting events.

[1] Laboratory of Genetics and Physiology, National Institute of Diabetes and Digestive and Kidney Diseases, US National Institutes of Health, Bethesda, Maryland 20892, USA. [2] Department of Biomedical Science and Engineering, Konkuk University, Seoul 05029, Republic of Korea. [3] Department of Cell and Developmental Biology & Dental Research Institute, Seoul National University, Seoul 110-749, Republic of Korea. [4] Department of Life Systems, Sookmyung Women's University, Seoul 140-742, Republic of Korea. [5] Transgenic Core, National Heart, Lung, and Blood Institute, US National Institutes of Health, Bethesda, Maryland 20892, USA. * These authors contributed equally to this work. ** These authors jointly supervised this work. Correspondence and requests for materials should be addressed to H.Y.S. (email: hayounshin@konkuk.ac.kr) or to L.H. (email: lotharh@mail.nih.gov).

The clustered regularly interspaced short palindromic repeat (CRISPR)-Cas9 system has been developed into an effective genome engineering tool[1–5]. CRISPR/Cas9 gene editing replaced previous gene-targeting technologies[6] and the direct injection of Cas9 and guide RNAs into zygotes provides an efficient and cost effective means to target the mouse genome[7]. The application of the CRISPR/Cas9 system in genome engineering requires two components, the single guide RNA (sgRNA) and the Cas9 nuclease[8]. Cas9 nuclease recognizes the protospacer adjacent motif (PAM) in the targeted region, which is adjacent to sgRNA, and creates double-strand breaks. Double-strand breaks are rapidly repaired by either non-homologous end joining (NHEJ) or homology-directed repair. NHEJ-mediated DNA repair often creates short deletions, occasionally large deletions, insertions and point mutations[7,9,10]. CRISPR/Cas9 has been used successfully to disrupt individual and multiple target genes and knock-in mice were generated to investigate biological functions[7,9,11–18]. Site-specific genomic modification has also provided new opportunities to interrogate the biological significance of transcriptional enhancers[19–24].

Although the CRISPR/Cas9 system has received vast attention, only a relatively small number of reports have described its application to introduce mutations in the mouse germline and the extent of molecular consequences has not been systematically explored. Our laboratory has targeted 17 genomic sites in the mouse genome and the respective mutations were analysed in 632 founders. Additional detailed analyses were performed in 54 lines established from specific founders. Specifically, we deleted sequences in enhancers bound by the transcription factor (TF) STAT5 (refs 19,25,26) and sequences recognized by CTCF (DOI: 10.1093/nar/gkx185), a protein known to aid in the establishment of functional chromatin loops. We investigated the molecular consequences on targeting sites with single sgRNAs and identified prevalent asymmetric deletions, preferred sites at which deletion occurs and large deletions. We also investigated strategies to delete juxtaposed sites and determined that only a sequential two-step, but not a one-step, targeting approach yielded reliable results. Our analyses permitted an assessment of target specificities, deletion efficiencies and size distributions based on one- and two-step targeting approaches.

## Results

**Asymmetric deletions using single sgRNAs.** CRISPR/Cas9 gene editing has been used successfully to target the mouse genome (Supplementary Table 1). To obtain in-depth knowledge on the extent of molecular consequences at target sites, we analysed deletions introduced at 17 loci in the mouse genome. By directly injecting the Cas9 machinery into mouse zygotes, we targeted five enhancers bound by the cytokine-sensing TF STAT5 in three genomic loci, Stat5a (A)[20], Socs2 (B)[26] and Wap (C)[19] (Fig. 1). We also targeted 10 sites bound by CCCTC-binding factor (CTCF), a DNA-binding protein proposed to participate in generating chromatin loops[27], in the Wap (D) and Csn (E) loci (DOI: 10.1093/nar/gkx185; Supplementary Fig. 1). Lastly, we targeted one enhancer bound by STAT5 and NFIB, a TF involved in epithelial cell differentiation[28–30] in the Csn (F) locus (Supplementary Fig. 1). These genomic sites were targeted individually or in combination. Juxtaposed sites within a given locus were targeted either simultaneously or successively, that is, in a one-step or two-step procedure (Fig. 1, Supplementary Notes 1–3 and Supplementary Table 2). Four distinct targeting strategies were pursued (detailed diagrams shown in Figs 2–4). We targeted individual TF-binding sites with only one corresponding sgRNA (Type 1), targeting individual TF-binding sites with more than one sgRNA (Type 2) and more than one TF-

**Figure 1 | Targeting 17 sites in the mouse genome with CRISPR/Cas9.** STAT5 TF-binding sites (GAS motif), CTCF-binding regions and an NFIB-binding site were targeted for deletion. STAT5-binding sites in three gene loci (A, Stat5 (ref. 20); B, Socs2 (ref. 26); C, Wap[19]) were targeted individually (A, B and C-1). The three STAT5-binding sites in the Wap super-enhancer[19] were deleted in four different combinations (C-1/2, C-1/3, C2/3 and C-1/2/3). Combined deletion of two or more STAT5-binding sites within the same gene locus was accomplished through a successive (two-steps) or simultaneous (one-step) targeting. A total of 11 CTCF-binding sites in two gene loci (D, Wap; E, Csn) were targeted individually and in different combinations. (F), an NFIB-binding site in the Csn locus was targeted (Supplementary Fig. 1). Target mutations were identified in 632 founder mice and 54 mutant lines were established. The positions of reference genes in the respective loci are indicated as coloured boxes. A, Stat5a; B, Socs2; C and D, Wap; E, Csn1s1; F, Csn3.

binding site with several sgRNAs (Type 3). In addition to targeting more than one site simultaneously (one-step, Type 3), we also targeted them successively (two-steps, Type 4). To accomplish this, we first targeted specific TF-binding sites and generated homozygous mutant mice, which subsequently served as hosts for targeting additional sites in the same gene locus. Molecular consequences of targeting events at 17 sites were investigated in more than 630 founder mice and 54 established lines, using both polymerase chain reaction (PCR) and DNA sequencing. The number of founders from each targeted site is shown in Supplementary Table 2. Deleted sequences from each individual founder are shown in Supplementary Notes 1–3. Each founder is the result of a distinct deletion induced by a given targeting event.

Cas9 nuclease recognizes the PAM, typically the NGG sequence adjacent to sgRNA in target DNA, and induces a double-strand break between the third and fourth nucleotides from PAM[31], but the orientation of deletions has not been reported. We examined the possibility of preferential orientations and distinguished between symmetric and asymmetric deletions (Fig. 5a). Only deletions obtained from injections of single sgRNAs were analysed, thereby avoiding effects of multiple variables (Fig. 2 and Supplementary Note 1). Deletions upstream of the Cas9-cutting site that were equal or less than 1.5-fold compared to downstream ones were defined as symmetric. Deletions exceeding 1.5-fold at either end were considered

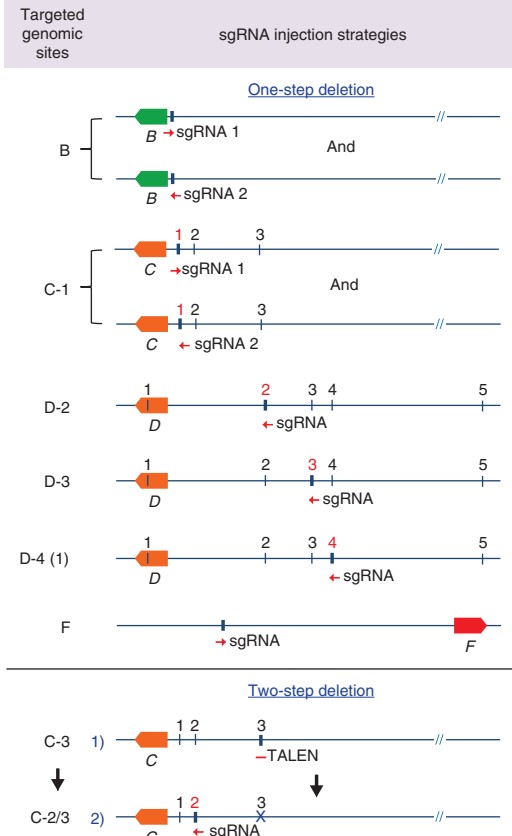

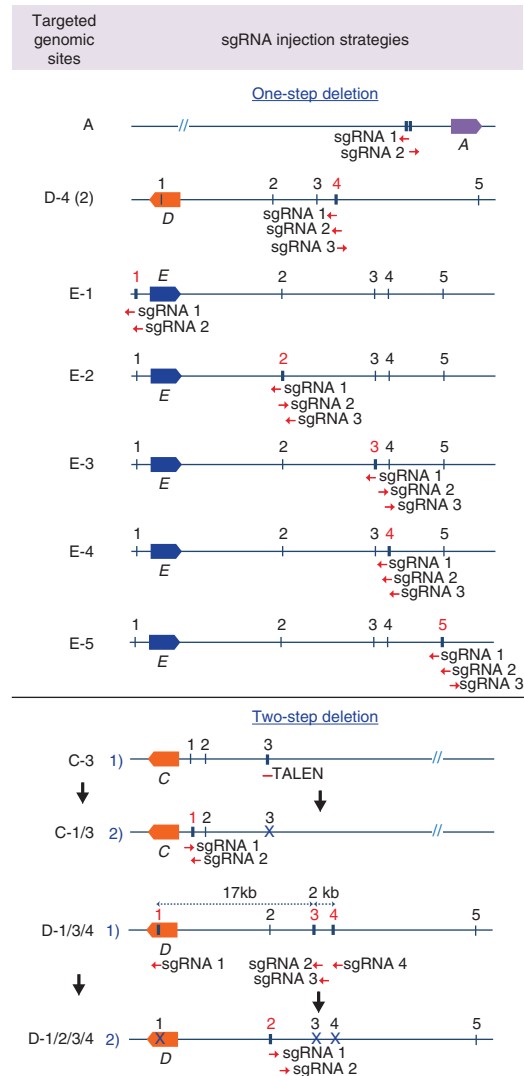

**Figure 2 | Targeting individual genomic sites with corresponding single sgRNAs.** Two genomic sites (B and C-1) were targeted independently with two individual sgRNAs (1 and 2). Four individual genomic sites (D-2, D-3, D-4(1) and F) were targeted with one sgRNA each. The deletion of two juxtaposed genomic sites (C-2/3) was generated in two steps. A single sgRNA-targeting site C-2 was injected into zygotes from mice carrying already a deletion in site C-3. The deletion of site C-3 was generated by TALEN[19]. B, *Socs2*; C and D, *Wap*; E, *Csn1s1*; F, *Csn3*. The red numbers refer to the sites being targeted in the respective experiments.

**Figure 3 | Targeting individual genomic sites with more than one sgRNA.** Each of seven individual genomic sites (A, D-4(2), E-1, E-2, E-3, E-4 and E-5) was simultaneously targeted with two or three sgRNAs. Combined deletions of more than one genomics site (C-1/3 and D-1/2/3/4) were accomplished in two steps. Two sgRNAs targeting site C-1 were simultaneously injected into zygotes from mice carrying already a deletion in site C-3 (ref. 19). Two sgRNAs targeting site D-2 were simultaneously injected into mice carrying already a deletion in sites C-1/3/4. A, *Stat5*; C and D, *Wap*; E, *Csn1s1*. The red numbers refer to the sites being targeted in the respective experiments.

asymmetric. More than 80% of the deletions detected in 139 founders obtained from targeting nine different sites were asymmetric and extended in either direction of the Cas9-cutting site (Fig. 5b). Notably, more than 70% of the deletions exceeded a two-fold difference. Asymmetric deletions were prevalent in founders from all nine genomic sites (from 50 to 100% of frequency), suggesting that this result was not linked to specific sgRNA sequences (Fig. 5c). Symmetric deletions were preferentially observed in small deletions of less than 10 bp (Fig. 5d). From all asymmetric deletions, 59% extended towards the 5′ end and 41% towards the 3′ end of the sgRNA (Fig. 5e,f). However, this was not statistically significant ($P = 0.6$). The 18 previously published studies studies that have targeted individual loci in the mouse genome with single sgRNAs have not specifically addressed the symmetry of deletions[10,11,17,18,31–44]. Deleted sequences were available from seven studies[11,18,31,34,39,43,44] (Supplementary Data 1), but only one[18] showed large enough data sets to permit a direct comparison with ours. Kim and colleagues[18] targeted two genomic sites based on 84 founders and with a cutoff of two-fold, 82% of the deletions were asymmetric compared to 73% in our study with 139 mice representing nine genomic sites (Supplementary Fig. 2a). With a cutoff of 1.5-fold, 89% of the deletions were asymmetric compared to 82% in our study. While

the frequency of asymmetric deletions was similar, the maximum deletion size was 585 bp in our study compared to 269 bp and the deletion sizes of top 50% are bigger in our study (Supplementary Fig. 2b).

**Deletions preferentially occur at repeat sequences.** We had noticed that with any given sgRNA 45% of the mutant founders carried apparently identical deletions, although mutations are supposed to be independent from each other. On detailed examination of such prevalent deletions, we determined that they frequently occurred at repeat sequences in targeted regions (Fig. 6a). Notably, single or duplicated units of repeat sequences were retained at the deletion site. More than 60% of mutant founders from one specific genomic site (D-4), a CTCF-binding site in the *Wap* locus, carried the exact same

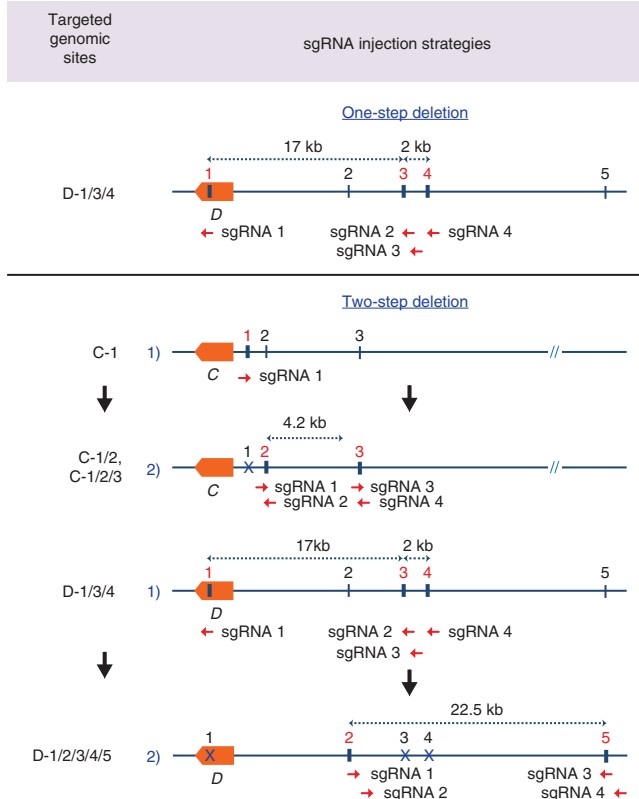

**Figure 4 | Targeting more than one genomic site with several sgRNAs.**
Three juxtaposed genomic sites (D-1/3/4) were targeted simultaneously with four sgRNAs. Combined deletion of more than one genomic site (C-1/2, C-1/2/3 and D-1/2/3/4/5) was accomplished in two steps. Four sgRNAs targeting sites C-2 and C-3 were simultaneously injected into zygotes from mice carrying already a deletion in site C-1 (ref. 19). Four sgRNAs targeting sites D-2 and D-5, which are 23 kb apart from each other, were simultaneously injected into mice carrying already deletions in sites C-1/3/4. C and D, *Wap*. The red numbers refer to the sites being targeted in the respective experiments.

deletion with the repeat sequences aligned at one end (Fig. 6b, upper panel). Over 30% of mutant founders from another genomic site (C-2/3), two STAT5-binding sites within the *Wap* super-enhancer[19], had the same repeat sequence aligned at both ends (Fig. 6b, bottom panel). Notably, 80% of deletions derived from one sgRNA (C-2/3) occurred at repeat sequences (Supplementary Fig. 3). Among the entire cohort of founders (56) carrying deletions at repeat sequences, 65% had deletions within a single copy of repeat sequences and 35% of founders had the duplication of repeat sequences (Fig. 6c). We exclusively analysed deletions obtained on targeting the genome with single sgRNAs (Fig. 2 and Supplementary Note 1). Deletions with repeat sequences aligned at one end were probably due to microhomology-mediated repair as previously reported *in vitro*[45–50] and *in vivo*[40,51–56] (Supplementary Table 1). However, to date the frequency of such deletion patterns had not been examined systematically in a large cohort and deletions with repeat sequences aligned at both ends had never been reported. Although the molecular mechanism that generates deletions with repeat sequences at both ends is unclear, microhomology-mediated end joining may facilitate the deletion with the repeat sequences aligned at one end[57] (Supplementary Fig. 4). These results indicate that CRISPR-based deletions do not simply occur randomly, but with preferential patterns.

**Large deletions created by single sgRNAs in zygotes.** Zhou et al.[11] have reported that injections of dual adjacent sgRNAs at a given site not only improved the deletion efficiency, but also increased the deletion size in nine founder mice. To further assess whether large deletions are also obtained on targeting single sites, we investigated the extent of deletions obtained on injection of one or more sgRNAs corresponding to a single genomic site (Fig. 7a). Although the median deletion size obtained with single sgRNAs (9 bp) (Fig. 2 and Supplementary Note 1) was shorter than that gained with more than one adjacent sgRNAs (84 bp) (Fig. 3 and Supplementary Note 2), we also observed large deletions of up to 600 bp with single sgRNAs (Fig. 7b). Notably, in one experiment the majority of founders (83%) exceeded the average deletion size (49 bp) and ∼40% harboured deletions over 200 bp (Supplementary Fig. 5). The deletion sizes generated by individual sgRNAs or more than one sgRNAs were independent of the guanine-cytosine (GC) content of the sgRNA or the distances between sgRNAs (Supplementary Fig. 6).

**Sequential versus simultaneous deletion of adjacent sequences.** Large deletions have been reported on simultaneously targeting more than one adjacent sites[7,9,13,14] (Supplementary Table 1). However, it is not clear if the deletion of juxtaposed sites can be achieved efficiently through the co-injection of the respective sgRNAs or whether a sequential deletion approach would be more robust. We addressed these questions and compared the deletion patterns obtained from mutant mice generated by simultaneous (one-step) or sequential (two-step) injection of sgRNAs covering seven sites (Fig. 8a, Fig. 4 and Supplementary Note 3). While the range of short deletions (<400 bp) obtained with both strategies was not significantly different (average deletion size of ∼8 bp per site and ∼39 bp per site), large deletions (>400 bp) of up to 24 kb were only obtained by co-targeting loci (Fig. 8b). Strikingly, among 45 founders obtained on simultaneously injecting sgRNAs for two sites more than 50% of the deletions were classified as large (Fig. 8c). Importantly, we initially failed to identify these large deletions due to the PCR screening strategy, which typically amplifies short fragments (∼400 bp) spanning individual sites. These large deletions were only detected using serial PCR primers spanning the entire target region.

Through in-depth sequence analysis, we identified two distinct deletion patterns, the 'stitched large deletion' and the 'continuous large deletion' (Supplementary Fig. 7). In one experiment, we observed a combination of short and large deletions (>2 kb) in a 7 kb region, which resulted in the stitched large deletions (Supplementary Fig. 7a). In another experiment, we also observed continuous large deletions over 20 kb in size that removed the entire sequence between sgRNAs. Strikingly, all nine founders from one particular experiment harboured these large deletions (Supplementary Fig. 7b and Supplementary Note 3). Based on the definition of microhomology-based deletions[45], we did not observe any microhomology-based large deletions with any given sgRNA injection method. Collectively, our results indicate that although the simultaneous targeting can rapidly generate deletions of multiple sites, it frequently creates large deletions, possibly removing potential regulatory elements. Although time consuming, two-step targeting appears to be the more reliable approach to precisely delete individual sites within a given locus.

**Insertions.** In addition to deletions, we also observed insertions (Supplementary Notes 1–3 and Supplementary Table 3). The frequency of insertions was 4% on targeting individual genomic sites with single sgRNAs, 10% on targeting individual genomic sites with more than one sgRNAs and 6% on simultaneously

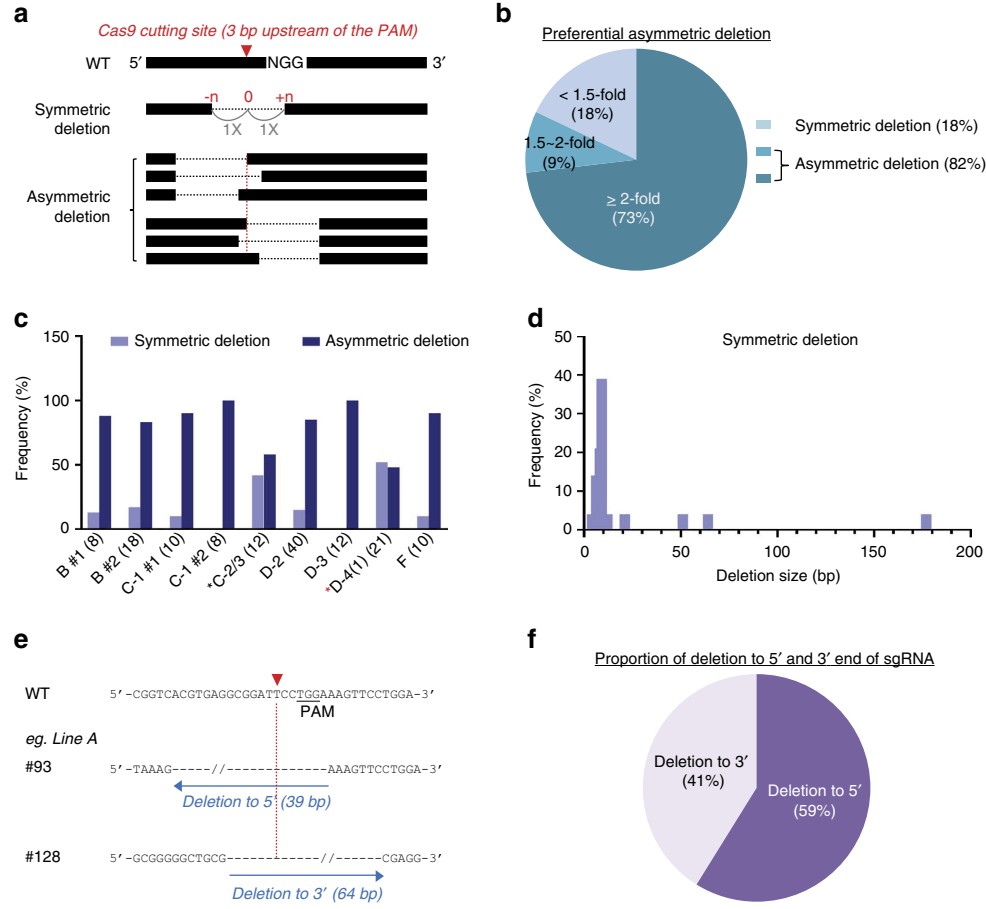

**Figure 5 | Asymmetric deletions. (a)** Schematic diagram of symmetric and asymmetric deletions detected in mice targeted with a single sgRNA. Red triangle, Cas9-cutting site three base pairs upstream of the PAM sequence. Symmetric deletions were defined as those with an equal or less than 1.5-fold ratio between the upstream and downstream Cas9-cutting site. In asymmetric deletions, the difference at either site was more than 1.5-fold than at the other site. **(b)** Percentage of symmetric and asymmetric deletions identified in CRISPR/Cas9-targeted mice ($n = 139$). More than 80% of deletions were asymmetric and more than 70% of the deletions exceeded a two-fold difference. Only deletions obtained from a single sgRNA injection were analysed to avoid the effect of multiple variables. Deletions obtained from a single sgRNA injection: deletions targeting TF-binding site B, C-1, C-2/3, D-2, D-3, D-4(1) and F. **(c)** Ratio of symmetric and asymmetric deletions obtained with each sgRNA. B #1, $n = 8$; B #2, $n = 18$; C-1 #1, $n = 10$; C-1 #2, $n = 8$; C-2/3, $n = 12$, D-2, $n = 40$; D-3, $n = 12$; D-4(1), $n = 21$; F, $n = 10$. Asterisk (*), sgRNAs with identical deletions identified in more than one half of the founders. **(d)** Frequency of symmetric deletions obtained with different deletion sizes. **(e)** Representative examples of deletions towards the 5′ end and 3′ end of sgRNA. If the deletion at the upstream Cas9-cutting site was longer ($\geq 1.5$-fold) than that at the downstream one, it was defined as a 5′ deletion and vice versa. **(f)** Percentage of deletions towards the 5′ end and 3′ end of sgRNA.

targeting more than one genomic site with several sgRNAs (Supplementary Table 3). We have also observed two different types of insertions, insertion combined with deletions (Type A) and insertion only (Type B) (Supplementary Notes 1–3). Although most insertions consisted of only a few nucleotides, we also observed a large insertion of 800 nucleotides comprised of repetitive sequences (Supplementary Note 3a).

**Avoiding genotyping pitfalls linked to large deletions.** Standard PCR genotyping methods are usually employed to screen for desired mutations in founder mice (F0). Initially, we designed PCR primers to examine short (400–500 bp) genomic regions surrounding targeted sites. However, the presence of large deletions in one allele with the other allele being wild-type can cause misleading PCR results as only the wild-type allele would be amplified. These mice would be incorrectly categorized as wild-type based on their apparent genotype, which failed to detect the mutant allele (Fig. 9a). Similarly, mice carrying deletions on both alleles could be misidentified. A regular PCR strategy would detect the small deletion on one allele but would miss a larger one

on the second allele that extends the location of one or both primers. These mice would misleadingly appear to be homozygous and the 'hidden deletion' would go unnoticed. Indeed, most of the 'homozygous' founders identified in our study were not genuine homozygous mutants but rather compound heterozygotes (Supplementary Note 3). Only the use of PCR spanning the entire targeted loci of up to 30 kb revealed the biallelic complexity of CRISPR/Cas-induced deletions. The presence of biallelic deletions of different sizes at multiple target sites was even more difficult to decipher (Fig. 9b). In our hands, 7 out of 30 founders were initially incorrectly categorized due to the large deletion (Supplementary Table 4). Lastly, when the large deletions were generated in intervening regions between two target sites, they were frequently missed if only the target sites were sequenced (Fig. 9c). Such large deletions could compromise the validity of biological studies and to avoid such problems, alternative genotyping methods, such as genomic qPCR[58] or even whole-genome sequencing, might be necessary.

To decode complex genotypes, especially large deletions obtained with two or more sgRNAs targeting juxtaposed sites,

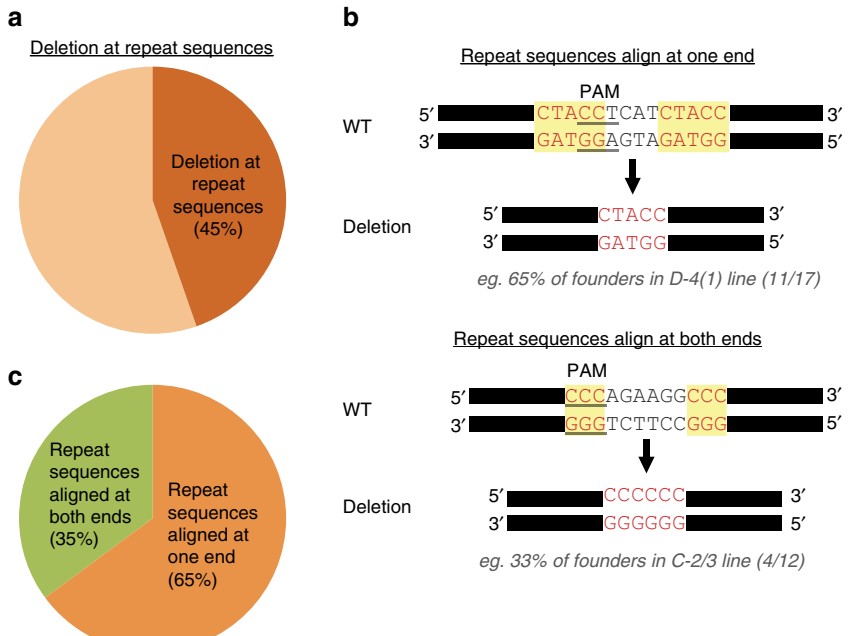

**Figure 6 | Preferential deletions at repeat sequences.** (**a**) Average frequency of deletions found at repeat sequences by single sgRNA injections. Repeat sequences were aligned in ~45% at one end or both ends (total $n = 122$; deletion at repeat sequences, $n = 56$). Only deletions obtained from injections with single sgRNAs were analysed to avoid effects of multiple variables. Deletions obtained from single sgRNA injections: Deletions targeting TF-binding site B, C-1, C-2/3, D-2, D-3, D-4(1) and F. (**b**) Representative examples of repeat sequences aligned at one end (upper panel) and both ends (bottom panel). More than 60% of mutant founder mice targeting the genomic site D-4 exhibited the exact same deletion that retained only a single copy of repeat sequence. More than 30% of mutant founder mice targeting the genomic site C-2/3 showed the exact same deletion and repeat sequences remained at both ends. (**c**) Percentage of repeat sequences aligned at one end and both ends in founder mice carrying deletions at repeat sequences.

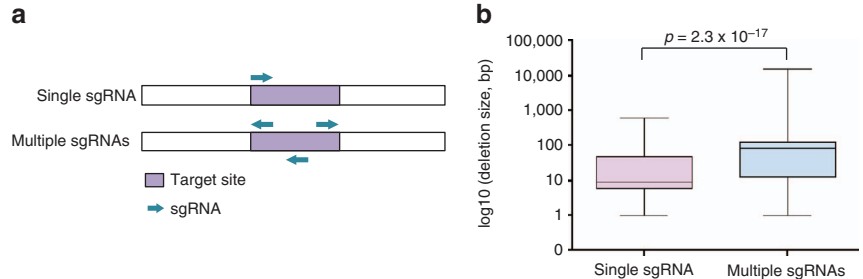

**Figure 7 | Large deletions obtained with single sgRNAs.** (**a**) Diagram of single and multiple sgRNAs targeting specific sites. Targeted sites are shown in purple and sgRNAs are indicated as cyan arrows. (**b**) Comparison of deletion sizes generated by single sgRNAs and multiple sgRNAs injected into mouse zygotes (total number of founder mice, $n = 243$; single sgRNA injection at a single site, $n = 122$; multiple sgRNA injections at a single site, $n = 121$). Single sgRNA injection at a single site, deletions obtained by targeting TF-binding site B, C-1, C-2/3, D-2, D-3, D-4(1) and F; multiple sgRNA injection at a single site, deletions obtained by targeting TF-binding site A, C-1/3, D-4(2), D-1/2/3/4, E-1, E-2, E-3, E-4 and E-5. The median deletion size obtained with single sgRNAs (9 bp) was smaller than that with multiple sgRNAs (84 bp). The deletion size generated with a single sgRNA was up to 600 bp. Median, middle bar inside the box; IQR, 50% of the data; whiskers, 1.5 times the IQR.

we used serial PCRs spanning sequences within loci as well as outside primers spanning entire loci (Supplementary Fig. 7c). Using this strategy, we easily identified deletions of more than 22 kb. In summary, the simultaneous sgRNA injections resulted in complex deletions and the generation of F1 mice is required to decode their exact genomic architecture. Moreover, simultaneous targeting of sites separated by up to 23 kb results in the deletion of the entire region. Thus, to restrict deletions to the desired sites we propose a sequential, two-step, targeting approach.

## Discussion

Although the CRISPR/Cas9 technology has been rapidly adopted as the premier genome editing tool for a range of organisms, much remains to be learned about the molecular consequences obtained on targeting the mouse genome, the collateral damage and the scars left behind. Based on more than 630 founder mice and 54 established lines covering 17 genomic sites, we have now acquired a more detailed understanding of deletion patterns obtained on injection of single or multiple sgRNAs into mouse zygotes. Preferential deletion patterns and large deletions were frequently obtained on targeting the mouse genome with single sgRNAs. Our studies also highlight that attempts to individually delete juxtaposed sites in a given locus through the co-injection of several sgRNAs almost exclusively results in large deletions spanning the entire sequence between the outside sgRNAs. This problem can be avoided by sequentially targeting sites, a time consuming, yet reliable, two-step process.

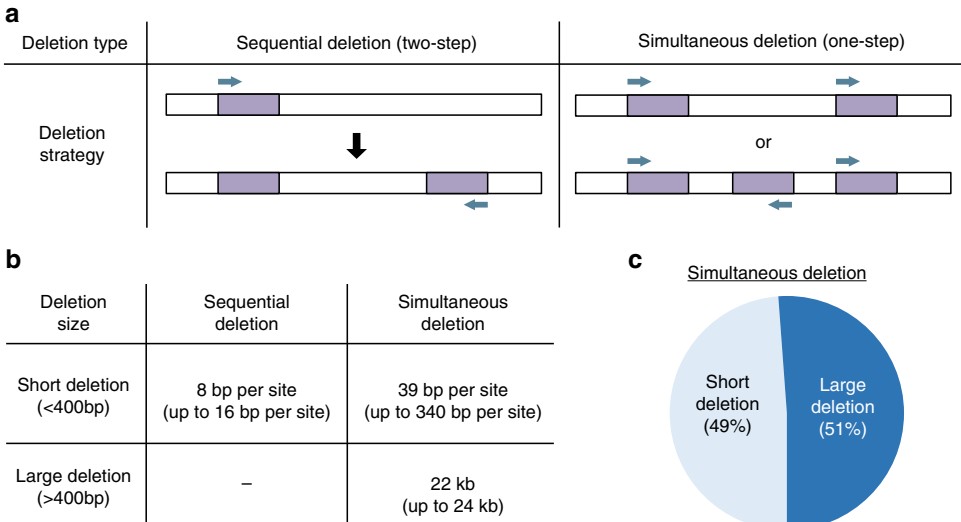

**Figure 8 | Deletion sizes obtained on sequentially (two-steps) and simultaneously (one-step) targeting the mouse genome. (a)** Schematic diagram of deleting several sites in a single gene locus using sequential or simultaneous sgRNA targeting. The targeted site is shown in purple and sgRNAs are indicated as cyan arrows. **(b)** Comparison of deletion sizes obtained from sequential and simultaneous sgRNA-targeting approaches. Sequential sgRNA injections, deletions obtained by targeting TF-binding site C-2/3; simultaneous sgRNA injections, deletions obtained by targeting TF-binding sites C-1/2, C-1/2/3, D-1/3/4 and D-1/2/3/4/5. Deletions smaller than 400 bp and identified by typical PCR genotyping method were called 'short deletion'. Those over 400 bp were considered 'large deletions'. Results are shown as the mean (total number of founder mice, $n = 65$; sequential deletion, $n = 20$; simultaneous deletion, $n = 45$). **(c)** The percentage of short and large deletions obtained from mutant mice generated by simultaneous injection with more than one sgRNA.

It has been suggested that CRISPR/Cas9-mediated DNA cleavage is randomly repaired by the NHEJ pathway[59]. However, we have observed hitherto unreported deletion patterns, which provided additional insight into the targeting accuracy. It has been reported that the seed sequence next to the PAM site are crucial for determining the target specificity[60] and Cas9 usually cleaves between the third and fourth nucleotides from PAM[31]. Our study demonstrated that the majority of CRISPR/Cas9-mediated deletions using single sgRNAs occur asymmetrically, independent of the targeted locus. We also analysed data from a study by Kim's laboratory[18] and found a similar preference of asymmetric deletions. These findings will help in predicting potential deletion sites and they may further aid sgRNA design to achieve targeted deletions at a high efficiency. Based on our findings that deletions frequently occur at repeat sequences a rationale design for sites with a high frequency of reproducible deletions should be possible.

When co-cutting a chromosome with two sgRNAs, the excised DNA fragment is occasionally inverted and re-integrated. Boroviak et al.[61] have elegantly demonstrated that such inversions happen frequently when two sgRNAs flanking large pieces of DNA (0.155–1.15 Mb) were co-injected into mouse zygotes. We have not specifically investigated the presence of inversions in mice generated by co-injecting two nearby sgRNAs. However, DNA sequencing analyses of PCR fragments spanning both cutting sites have not revealed any inversion of the intervening sequences. Possibly, the short DNA pieces excised in our experiments were quickly degraded by exonucleases and therefore they do not have a chance to be re-integrated into the chromosomes as the large pieces do. In one experiment, we deliberately deleted a 22 kb fragment by co-injecting sgRNAs, and all nine founder mice were apparently homozygous. Only analyses on the F1 generation determined that these mice were compound heterozygous, with the two alleles carrying slightly different deletions.

We have observed large deletions of up to 600 bp induced by single sgRNAs. We also obtained deletions of up to 24 kb induced by multiple sgRNAs targeting more than one juxtaposed site, which is in agreement with other studies[62]. Large deletions are easily overlooked in conventional PCR screening strategies and the observation that the two alleles can routinely harbour deletions of distinctly different sizes further complicates their identification. Based on our analyses, we suggest that true molecular changes can only be identified in the F1 generation. Moreover, mosaic mutations offer additional challenges that need to be sorted out in the F1 generation. To rapidly obtain homozygous mutants, apparent identical homozygous founders can be bred to each other. However, the possible misinterpretation of founder genotypes will yield undesirable compound heterozygosity. Thus, it would be prudent to rely only on genotypes from the F1 generation obtained by breeding founders with wild-type mice. Large deletions at target sites and even at some distance require additional analyses, possibly whole-genome sequencing, such as those reported to screen the off-target sites[63].

## Methods

**Mice.** CRISPR/Cas9-targeted founder mice were obtained from the transgenic core of the National Heart, Lung, and Blood Institute. Eight-week-old C57BL/6 mice were purchased from Charles River and bred with founder mice to segregate the mosaicism. All animal procedures were followed by the National Institutes of Health, National Institute of Diabetes, Digestive and Kidney Diseases guidelines for the care and use of laboratory animals.

**Design of CRISPR sgRNAs and microinjection into mouse zygotes.** The CRISPR sgRNAs were designed based on the nearest PAM of the target sequence and their off-target scores were evaluated by the online tool at crispr.mit.edu[19,20]. Each sgRNAs were cloned into the pDR274 plasmid vector (Addgene #42250), and in vitro transcribed using the MEGAshortscript T7 kit (Life Technologies). Cas9 mRNA was in vitro synthesized from the MLM3613 plasmid vector (Addgene #42251) using the mMESSAGE mMACHINE T7 kit (Life Technologies). Cas9 mRNA (100 ng µl$^{-1}$) and sgRNAs (50 ng µl$^{-1}$) were microinjected into the cytoplasm of fertilized eggs of superovulated B6CBAF1/J female mice (JAX) and implanted into oviducts of pseudopregnant fosters (Swiss Webster, Taconic Farm). To target individual genomic sites, one or more than one sgRNA (up to three sgRNAs) were designed and injected independently or simultaneously. To target more than one genomic site, several sgRNAs (up to a total of four sgRNAs, two

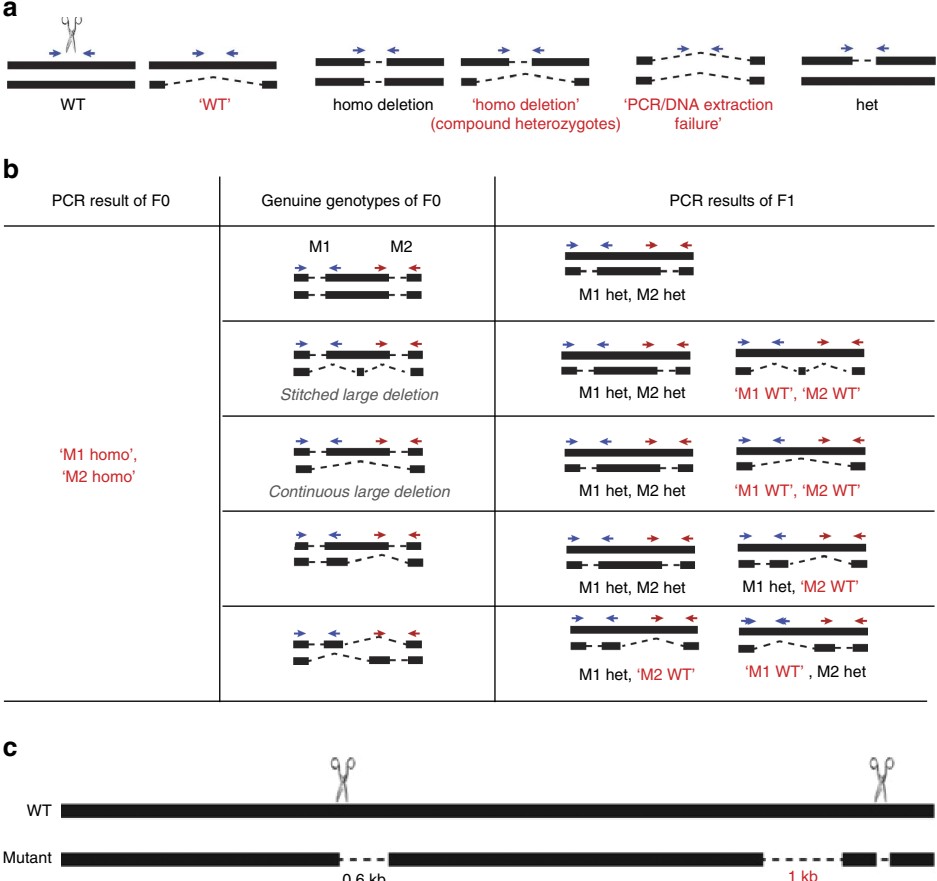

**Figure 9 | Screening for compound heterozygote deletions that are disguised as homozygote deletions.** (**a**) Diagrams depict misleading genotyping results caused by large deletions at one of the two alleles. Misleading results are shown in red with quotation marks. All possible genotypes generated from single site mutagenesis are shown. (**b**) Diagrams depict examples of potential misleading genotyping results caused by stitched large deletions and continuous large deletions. Blue and red arrows indicate PCR primers. Misleading results are shown in red with quotation marks. WT, wild-type; het, heterozygous; homo, homozygous; M1: mutated site 1; M2: mutated site 2. (**c**) Schematic demonstration of a mutant derived from simultaneous injection of two sgRNAs, which carries a 1 kb deletion between two targeting sites in addition to the desired short deletions at the target sites.

sgRNAs for each site) were injected simultaneously or sequentially. When we injected two sgRNAs, we normally used $50\,ng\,\mu l^{-1}$ for each sgRNA. When more than two sgRNAs are injected, we reduced the concentration proportionally, so that the total sgRNA concentration does not exceed $100\,ng\,\mu l^{-1}$ (that is, $25\,ng\,\mu l^{-1}$ for each sgRNA was used when four sgRNAs are co-injected). Some sgRNAs caused lethality. When that happens, we normally diluted both Cas9 mRNA and sgRNA (first by 4-folds and then by 10-folds), which often improves litter sizes. The detailed targeting strategies for individual genomic sites were shown in Supplementary Notes 1, 2 and 3. The number of sgRNAs injected into the individual genomic sites: (1) Targeting individual genomic sites with single sgRNAs (Site B and C-1, two independent sgRNAs; site C-2/3, D-2, D-3, D-4(1), F, one sgRNA), (2) Targeting individual genomic sites with more than one sgRNA (Site A, C-1/3, D-1/2/3/4, E-1, two sgRNAs simultaneously; site D-4(2), E-2, E-3, E-4, E-5, three sgRNAs simultaneously) and (3) Targeting more than one genomic site with several sgRNAs (Site C-1/2, C-1/2/3, D-1/3/4, D1/2/3/4/5, total four sgRNAs simultaneously).

**Generation of CRISPR mouse lines and genotyping.** Founder mice were bred with wild-type mice to obtain heterozygous F1 mice. F1 mice with an identical genotype were interbred to generate F2 homozygous mice. All mice were genotyped by PCR amplification of genomic DNA isolated from the tip of the tail, followed by Sanger sequencing. Large deletions were identified by serial PCR genotyping using primers that were designed to amplify ∼400 bp encompassing the target sequence or long-range PCR.

**Statistical analyses.** All samples used for statistical analyses were randomly selected, and blinding was not applied. Before any statistical analysis, a Shapiro–Wilk normality test was applied to validate the normal distribution of the data. Significance for box plots was determined using Wilcoxon signed-rank test.

A binomial test was employed to examine the statistical significance of prevalent orientation of deletions.

**Data availability.** The authors declare that the data supporting the findings of this study are available within the article and its Supplementary Information files or from the corresponding author on reasonable request.

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

## Acknowledgements

This research was funded by the IPR of the NIDDK/NIH and by a grant of the Korean Health Technology R&D Project, Ministry of Health & Welfare, Republic of Korea (HI15C1184) to H.K.L.

## Author contributions

H.Y.S. designed, executed and supervised the genotyping, identified and validated founder mice, analysed data and wrote the manuscript. C.W. designed, conducted and supervised the genotyping, analysed data and wrote the manuscript. H.K.L. performed the genotyping, established lines, analysed data and wrote the manuscript. K.H.Y. designed, performed and supervised the genotyping, identified founder mice and analysed data. X.Z., T.K., C.M.Y. and T.M. performed genotyping, identified founder mice, established lines and analysed data. C.L. conceived idea and generated all founder mice. L.H. conceived and supervised the study, analysed data and wrote the manuscript. H.Y.S. and L.H. wrote and finalized the manuscript, and all authors reviewed and approved the submitted version.
