## [Peer Review File · Nature Communications]

Reviewer #1 (Remarks to the Author)

Review of "CRISPR/Cas9 biased mutagenesis in the mouse genome"

In this manuscript, Shin and colleagues report the result of the analysis of the consequences of CRISPR-Cas9-mediated mutagenesis in the mouse zygote.

By examining the genetic 'scars' left by guideRNAs designed to target 20 distinct sites in over 500 founder mice, they report that deletions are often asymmetric with respect to the cleavage site and extend preferentially in the 5' direction. They also report that while co-injection of multiple of gRNAs targeting nearby sites tend to generate, on average and as expected, larger deletions, even a single guideRNA can induce deletions of up to several hundred nucleotides in a substantial number of zygotes. They correctly point out that these larger deletions are likely to be missed by conventional PCR-based screening approaches. Finally, they show that sequential, rather than simultaneous, gRNA injection is preferable for multiplex mutagenesis, as it reduces the frequency of undesired large deletions spanning the regions between the gRNAs target sites.

The experiments are generally well designed, and the conclusions are consistent with their results, as far as one can tell from the limited data shown. Overall, this work could be of interest to the scientific community, given the growing interest in the generation of genetically engineered mice using CRISPR-based methods.

However, the general validity of the results is hard to judge since the authors only present a very superficial description of them. More specifically:

a) characterization of deletions symmetry. The authors should clearly state whether only deletions of three nt or more are considered for this analysis, a deletion of one or three nt have to be asymmetric by default and as these mutations are particularly common they would inflate the results. Unfortunately figure 1 only shows a schematic and a pie chart with the total numbers. Much more useful would have been a plot showing symmetry ratios of deletions of different size.

Also, and this is true throughout the manuscript, the authors should report the results separately for each of guideRNA, as it is likely that different sites will have different propensity at generating symmetric or asymmetric deletions.

A table and a couple of scatter plots would provide these essential information.

b) The same problem applies to figure 2, the reported results are limited to founders generated with only two of the many sgRNAs used in the experiments. Representative examples are useful, but the entire dataset is essential for the reader. I would strongly recommend that, in addition to adding details and information to the main figure, a full list of all mutations generated by each gRNA used is included as supplementary dataset.

c) Figure 3 suffers of the same limitations. Only aggregate results are shown and in figure b and it is impossible to know what the variability between different gRNAs and between different injection of the same gRNA are.

d) When using 2 gRNAs targeting the same region, how frequently did the authors observe inversions or tandem duplications of the intervening region? This is another important result that should be reported.

In conclusion, without a careful statistical analysis and a full description of the results it is difficult to appreciate the relevance of the findings reported by this otherwise potentially interesting work.

Additional comments:

Supplementary figure 1 should be included as main figure. the location of the various gRNAs used

should be indicated on the schematics. The figure legend should be corrected ("Color A..?"). In general, some editing by a native English speaker would improve clarity throughout the manuscript.

I am not sure I understand what the authors refer to when they use the term 'locus' in this manuscript.

Reviewer #2 (Remarks to the Author)

Shin et al. CRISPR/Cas biased mutagenesis in the mouse genome

The authors study the outcome of CRISPR/Cas assisted gene editing in mouse zygotes at a large number of genomic sites with special attention to the presence of yet unexpected mutagenesis events at the target loci. Characterizing small deletions at single sites they describe a bias in the size of deleted sequences towards the 5' - end of the sgRNA and occasionally repair products likely generated via microhomology mediated end joining. Careful genotyping of alleles generated by use of two or more sgRNAs revealed the presence of unexpected large deletions and deletion patterns which cannot be detected by plain PCR coverage of just a short genomic sequence.

The work of Shin et al. is the first study on the presence and frequency of undesired/unexpected outcomes of CRISPR induced gene editing at the intended target genes in mouse zygotes, surveying a large number of target sites and founder mutants. The existence of such alleles has been so far largely ignored or such events were overlooked by use of PCR assays which detect only a small segment of the target gene. Therefore, the study of Shin et al. will be relevant for the community using CRISPR for gene editing in the mouse as well as other mammals. While the results of Shin do not disqualify CRISPR as an efficient mutagenesis approach they underline the importance of a carefully planned genotyping strategy covering the entire target gene.

Minor topics

1. In the section on repair at repeat sequences I would recommend to mention that microhomology mediated end joining is well recognized as an alt-NHEJ pathway, which has previously been found to bias repair products at target sites harboring short repeats. See e.g. the work of Bae et al. , *nature methods* VOL.11 NO.7 JULY 2014, 705

2. In the section on 'Misleading genotypes linked to large deletions..' I would prefer to include a note on the actual frequency of these events in the founders studied. Line 174 cites them as high frequency but the real numbers are missing.

Reviewer #3 (Remarks to the Author)

This manuscript by Shin et al. promises to reveal novel characteristics of Cas9-induced mutations resulting from mouse embryo injections. In reality, it falls far short of this goal. In describing their results, the authors provide essentially no primary, or even secondary, data, relying instead on high-level summaries. This makes it impossible to evaluate their conclusions. For example:

1. Their classifications of deletions often seem arbitrary – e.g., calling an event asymmetric only if the ratio of deletion sizes on the two sides of the expected cut site were > 1.5-fold.
2. The sequences of the target sites are not provided, so the work could not be repeated.
3. Because they are intended to represent multiple loci, the figures have no scale and are thus not very useful.
4. The actual numbers of embryos injected for each sgRNA and the number of independent events and their frequencies are not provided.
5. The sequences of all the characterized junctions should be provided. Since some were recovered multiple times, this should not be an unwieldy list if included in the Supplementary Information.

The authors also betray a lack of familiarity with the existing literature on Cas9-induced (and ZFN- and TALEN-induced) indels.

6. The preferred use of microhomologies flanking the cut site, as described in Figure 2a and Supplementary Figure 2, has been known for many years. A particularly clear example is Bae et al. (2014) *Nature Methods* 11: 705, but the phenomenon has been recognized for decades.

7. The junction described in Figure 2a has no basis in mechanism, unless there is something revealing in surrounding sequences not shown.

8. It has long been recognized that the PCR techniques commonly used to assess indels will underestimate the number of large deletions; nonetheless, very large deletions have been reported.

9. The generation of long deletions by simultaneous introduction of two nucleases targeting the same DNA segment has also been reported multiple times, for Cas9 and for ZFNs and TALENs.

10. The existence of asymmetric deletions has also been noted, although I am not aware of a systematic study. The authors might attempt to compare their findings with published results for other systems. For example, are the indels produced in embryos significantly different from those recovered from cultured cells?

The one useful suggestion in the manuscript is that researchers should be more careful about looking for the existence of cryptic long deletions, instead of characterizing cases with one detected indel as homozygous.

The data accumulated in this study could be welcome, if it were provided in a form that allows analysis by other researchers and if it were systematically related to previous results.

Response to reviewers

General comments

We have added all primary data as requested by the reviewers and have also included an additional 40 founders in which a new site (Csn super-enhancer) had been targeted. Thus this study now contains data from 24 loci, 632 founders and more than 54 lines.

Reviewer #1

The experiments are generally well designed, and the conclusions are consistent with their results, as far as one can tell from the limited data shown. Overall, this work could be of interest to the scientific community, given the growing interest in the generation of genetically engineered mice using CRISPR-based methods. However, the general validity of the results is hard to judge since the authors only present a very superficial description of them. More specifically:

a) Characterization of deletions symmetry. The authors should clearly state whether only deletions of three nt or more are considered for this analysis, a deletion of one or three nt have to be asymmetric by default and as these mutations are particularly common they would inflate the results. Unfortunately figure 1 only shows a schematic and a pie chart with the total numbers. Much more useful would have been or a plot showing symmetry ratios of deletions of different size.

Response

We included all deletion sizes we obtained from single sgRNA injections (from 1 bp to 585 bp). In our study, one or three nucleotide deletions were not common (1 nt: 9% with 3 out of 9 sgRNAs, 3 nt: 2% in 2 out of 9 sgRNAs, Supplementary Table 2) and therefore did not inflate the overall results. Even if we only consider deletions exceeding 3 nucleotides, the asymmetric deletions were prevalent (symmetric: 23%, asymmetric: 77%). As the reviewer suggested, we included the frequency of symmetric deletions by different size in Figure 2d.

Reviewer

Also, and this is true throughout the manuscript, the authors should report the results separately for each of guideRNA, as it is likely that different sites will have different propensity at generating symmetric or asymmetric deletions.

Response

Yes, we agree with reviewer's comment. We have now included (Figure 2c) the ratio of symmetric and asymmetric deletions for each sgRNAs (total 9 different sgRNAs and total 139 deletions). The frequency of asymmetric deletions was also predominant with each sgRNAs except sgRNA targeting C-2/3 and D-4(1) which generated many identical deletions (Supplementary Table 2).

Reviewer

b) The same problem applies to figure 2, the reported results are limited to founders generated with only two of the many sgRNAs used in the experiments. Representative examples are useful, but the entire dataset is essential for the reader. I would strongly recommend that, in addition to adding details and information to the main figure, a full list of all mutations generated by each gRNA used is included as supplementary dataset.

Response

As requested we have analyzed the entire dataset of single sgRNA injections (total 122 deletions) and show the average frequency of 'Deletion at repeat sequences' (45%) in Figure 3a. Through more detailed analyses, we determined the frequency of deletions at repeat sequences induced by each sgRNAs (total 9 different sgRNAs) in Supplementary Figure 1 and also showed the ratio of 'Repeat sequence aligned at one end' and 'Repeat sequence aligned at both ends' in Figure 3c. As the reviewer suggested, we included a full list of all mutations generated by each sgRNA in Supplementary Table 2.

Reviewer

c) Figure 3 suffers of the same limitations. Only aggregate results are shown and in figure b and it is impossible to know what the variability between different gRNAs and between different injection of the same gRNA are.

Response

We now provide a full list of all mutations with detailed information of each sgRNAs in Supplementary Table 2 (injection of single sgRNAs) and Supplementary Table 3 (Co-injection of several sgRNA targeting a single site). To determine whether the variability between different sgRNAs affected the deletion size, we analyzed the deletion sizes by the GC content of each sgRNAs and deletion sizes by the distance between each sgRNAs. These factors did not influence the deletion size and results are shown in Supplementary Fig 4. When we designed sgRNAs, we only used the 40-60% of GC content which is known to be optimal. This might be the one reason why the different GC content did not affect the deletion size in our case. We used the same injection method for all sgRNAs.

Reviewer

d) When using 2 gRNAs targeting the same region, how frequently did the authors observe inversions or tandem duplications of the intervening region? This is another important result that should be reported.

Response

Several papers have reported inversions and tandem duplications (PMID: 25660031 and 26742453). However, we did not detect inversion or tandem duplications in our founder genotyping results. Since many founders were mosaic and we only fully analyzed sequences from founders with clear deletions, we might have missed the existence of inversion or tandem duplications hidden in different alleles of mosaic founders.

Reviewer

In conclusion, without a careful statistical analysis and a full description of the results it is difficult to appreciate the relevance of the findings reported by this otherwise potentially interesting work.

Response

We have now fully analyzed our extensive data sets and the findings should aid scientists in planning experiments in mice.

Additional comments:

Supplementary figure 1 should be included as main figure. The location of the various gRNAs used should be indicated on the schematics. The figure legend should be corrected ("Color A.."?). In general, some editing by a native english speaker would improve clarity throughout the manuscript.

Response

We moved Supplementary Figure 1 to main Figure 1 and the location of each sgRNAs was indicated in the schematics. The figure legend was corrected.

Reviewer

I am not sure I understand what the authors refer to when they use the term 'locus' in this manuscript.

Response

We used the term 'locus' to refer to the genes and their regulatory regions that were targeted using sgRNAs. For clarity reasons we have provided the genomic coordinates.

Reviewer #2

Shin et al. CRISPR/Cas biased mutagenesis in the mouse genome
The authors study the outcome of CRISPR/Cas assisted gene editing in mouse zygotes at a large number of genomic sites with special attention to the presence of yet unexpected mutagenesis events at the target loci. Characterizing small deletions at

single sites they describe a bias in the size of deleted sequences towards the 5`- end of the sgRNA and occasionally repair products likely generated via microhomology mediated end joining. Careful genotyping of alleles generated by use of two or more sgRNAs revealed the presence of unexpected large deletions and deletion pattern which cannot be detected by plain PCR coverage of just a short genomic sequence. The work of Shin et al. is the first study on the presence and frequency of undesired/unexpected outcomes of CRISPR induced gene editing at the intended target genes in mouse zygotes, surveying a large number of target sites and founder mutants. The existence of such alleles has been so far largely ignored or such events were overlooked by use of PCR assays which detect only a small segment of the target gene. Therefore, the study of Shin et al. will be relevant for the community using CRISPR for gene editing in the mouse as well as other mammals. While the results of Shin do not disqualify CRISPR as an efficient mutagenesis approach they underline the importance of a carefully planned genotyping ideally strategy covering the entire target gene.

Minor topics

Reviewer

In the section on repair at repeat sequences I would recommend to mention that microhomology mediated end joining is well recognized as an alt-NHEJ pathway, which has previously be found to bias repair products at target sites harboring short repeats. See e.g. the work of Bae et al. , nature methods VOL.11 NO.7 JULY 2014, 705

Response

We have cited the paper of microhomology-based deletions found in the in vitro cell cultures (Bae et al, PMID 24972169).

Reviewer

In the section on `Misleading genotypes linked to large deletions..' I would prefer to include a note on the actual frequency of these events in the founders studied. Line 174 cirtes them as high frequency but the real numbers are missing.

Response

As the reviewer suggested, we included the actual frequency of misleading genotypes linked to large deletions (or the frequency of mosaic large deletions) in Supplementary Table 6 and the detail information in Supplementary Table 4.

Reviewer #3

This manuscript by Shin et al. promises to reveal novel characteristics of Cas9-induced mutations resulting from mouse embryo injections. In reality, it falls far short of this goal. In describing their results, the authors provide essentially no primary, or even secondary, data, relying instead on high-level summaries. This makes it impossible to evaluate their conclusions. For example:

Reviewer

Their classifications of deletions often seem arbitrary – e.g., calling an event asymmetric only if the ratio of deletion sizes on the two sides of the expected cut site were > 1.5 -fold.

Response

We agree with the reviewer's comments. In addition to the 1.5-fold cutoff we have now added a more stringent cutoff of asymmetric deletions (> 2 -fold) and 73% of the deletions are asymmetric. Results of the frequency of asymmetric deletions with both > 1.5 cutoff (82%) and > 2 -fold cutoff (73%) are now shown in Figure 2b.

Reviewer

The sequences of the target sites are not provided, so the work could not be repeated. Because they are intended to represent multiple loci, the figures have no scale and are thus not very useful.

Response

We now provide the sequences of the target sites for the entire dataset in Supplementary Tables 2, 3, and 4 and we indicated the scale of each target sites.

Reviewer

The actual numbers of embryos injected for each sgRNA and the number of independent events and their frequencies are not provided.

Response

As requested, we now include the number of injected embryos, the number of pups born, and mutation frequencies for each mutant line in Supplementary Table 5.

Reviewer

The sequences of all the characterized junctions should be provided. Since some were recovered multiple times, this should not be an unwieldy list if included in the Supplementary Information.

Response

We now provide the sequences of all characterized junctions in Supplementary Tables 2, 3, and 4.

Reviewer

The authors also betray a lack of familiarity with the existing literature on Cas9-induced (and ZFN- and TALEN-induced) indels.

Response

We agree with reviewer that the existing literature had not been adequately covered and we conducted a thorough literature search on CRISPR-induced and ZFN- and TALEN-induced indels and provided the relevant reference list in the Supplementary Table 1. Although several papers (approximately 8) mentioned the microhomology-based deletions which correspond to our 'Deletions with repeat sequences aligned at one end', these were mainly found in vitro cell culture systems. Although several papers have reported large deletions by targeting multiple sites, our study is the first to systemically analyze large deletions obtained from both single sgRNA injections and co-injection of

several sgRNAs in mouse zygotes and using large cohorts (632 founder mice). Preferential 5' asymmetric deletions and 'Deletions with the repeat sequences aligned at both ends' have never been reported in other systems.

Reviewer

The preferred use of microhomologies flanking the cut site, as described in Figure 2a and Supplementary Figure 2, has been known for many years. A particularly clear example is Bae et al. (2014) Nature Methods 11: 705, but the phenomenon has been recognized for decades.

Response

Yes, Bae et al identified the microhomology-based deletion using cell lines and we cited their study. Another study mentioned the microhomology-based deletion in vivo (PMID 26887046), but these investigators only showed 10 examples and did not perform any systematic analyses. Using in vivo mouse system, we performed systematic analyses on 'Deletion at repeat sequences' in large cohorts (total 122 deletions) and found that this deletion pattern is prevalent (average of 45%). We also found the existence of 'Deletions with repeat sequences aligned at both ends' which has never been reported in any other system, not just the 'Deletions with repeat sequences aligned at one end (or microhomology-based deletions).

Reviewer

The junction described in Figure 2a has no basis in mechanism, unless there is something revealing in surrounding sequences not shown.

Response

We cited the paper that shows the mechanism of microhomology-mediated end joining (PMID 18809224) and proposed the molecular mechanism based on our case in Supplementary Figure 2.

Reviewer

It has long been recognized that the PCR techniques commonly used to assess indels will underestimate the number of large deletions; nonetheless, very large deletions have been reported.

Response

As stated by reviewer, typical PCR genotyping method often underestimates the number of large deletions. Failure to identify large deletions in mosaic founder mice will result in misleading genotypes and a waste of time and resources, especially if there is a high frequency of large deletions (eg. Line D-1/2/3/4/5 in Supplementary Table 4). Large deletions are also not that useful for accurate functional studies. To avoid such waste, alternative genotyping method such as genomic qPCR (PMID 26755636) or even whole genome sequencing might be necessary.

Reviewer

The generation of long deletions by simultaneous introduction of two nucleases targeting the same DNA segment has also been reported multiple times, for Cas9 and for ZFNs and TALENs.

Response

Yes, several studies have reported CRISPR, TALEN, and ZFN generate large deletions by simultaneously targeting multiple regions both in vitro cell cultures and in vivo mouse. Although large deletions have been reported in CRISPR-mediated germ line manipulations in the mouse, there had been, to the best of our knowledge, no systematic analysis. Our study included a systematic analysis of large deletions in a cohort of over 630 founder mice and we have identified large deletions not only in mice obtained from simultaneously targeting several sites, but also from the single sgRNA injections. Notably, all founders (10) from one specific mutant line carry large deletions of up to 24 kb (Supplementary Table 4).

Reviewer

The existence of asymmetric deletions has also been noted, although I am not aware of a systematic study. The authors might attempt to compare their findings with published results for other systems. For example, are the indels produced in embryos significantly different from those recovered from cultured cells?

Response

We did the through literature search for papers describing asymmetric or symmetric deletions in mice, not just in CRISPR-induced deletions but also in other systems (ZFN and TALEN). However, we were unable to identify any papers specifically mentioning symmetric or asymmetric deletions. One study used the asymmetric donor DNA to create knock-in cells (PMID 26789497), but it was done in cell lines and they did not conduct a systematic analysis.

Reviewer

The one useful suggestion in the manuscript is that researchers should be more careful about looking for the existence of cryptic long deletions, instead of characterizing cases with one detected indel as homozygous.

Response

As suggested by the reviewer, we included the frequency of cryptic large deletions from mosaic founders in Supplementary Table 6.

Reviewer

The data accumulated in this study could be welcome, if it were provided in a form that allows analysis by other researchers and if it were systematically related to previous results.

Response

We now provide the entire set of mutated sequences in all founders and lines investigated in Supplementary Tables 2, 3, and 4. This should other researchers to conduct further analyses.

Reviewer #3 (Remarks to the Author)

This revised manuscript presents more data on experimental protocols and mutant sequences, but I had a very hard time determining what was actually done. For example, nowhere in the text do the authors say that they have used multiple sgRNAs for many of the individual sites shown in Figure 1. This information was only available in the supplementary materials. That made it difficult to understand the difference between “several sgRNAs targeting a single site” and “sgRNAs targeting several sites.” Define a “site.” Throughout the manuscript, the authors should state whether they are analyzing only independent mutations – i.e., ones derived from separate founders. Was sequencing done directly on founders, or only on F1 or F2 animals, as suggested in the Methods section?

The nomenclature for experiments listed in Table S5 is very obscure. What are the “Mutant Lines” into which injections were done? After much effort, I gather that Mutant Line B was injected separately with sgRNAs for site B (cf Fig. 1), but was the recipient already mutant? Information on Line C-2/3 in Table S2 suggests that embryos already mutant at site C-3 were injected with sgRNA #1 for site C-2. What about D-1/2/3/4? Was it already mutant? For what? And what sgRNAs were injected? Similarly for D-1/3/4, for example. Which sites were targeted, and what was the background? In Figure 4b, are these data for multiple sgRNAs targeting a single site, or multiple sites? How far apart are the sites?

In my view the authors continue to exaggerate the novelty of their observations. There are many more examples of microhomology-mediated deletions than are shown in Table S1a. No attempt has been made to compare current and prior results in any detail, perhaps in a scatter plot of deletion sizes. Such a plot for asymmetric deletions would have been welcome in Figure 2, beside that for “symmetric deletions.” Asymmetric deletions have been observed repeatedly as well, and a specific comparison with current experiments is needed. The definition of what is asymmetric continues to be arbitrary. The bias toward 5'-directed deletions has also been reported, and the bias shown here – 59% vs. 41% – is not large enough to have a strong effect on target choice. The value of making mutations at individual sites with sequential injections to avoid very large deletions seems obvious.

Some other information is missing. The text should mention that insertions were found at the cut sites as well, perhaps just giving their frequency and any particularly notable features. In the cases of large deletions, don't some of the junctions look like they are microhomology-based?

There are some errors to be corrected – e.g., the continuous large deletions illustrated in Fig. S5 appear to come from D-1/2/3/4/5 (see Table S4), not from E-1/2/3/4/5 as stated. On p. 3, it is not made clear the PAM is outside the sequence complementary to the sgRNA. On p. 6, it says the longest deletion from a single sgRNA was 6 kb, while a maximum of 0.6 kb is given elsewhere.

Overall, this paper provides insufficient information on how experiments were done and which experiments are described in each figure or in the text.

We thank the reviewer for providing detailed comments and feedback. We have now added additional information and figures to clarify all points. We believe that this comprehensive study of more than 630 founder mice and more than 50 lines covering 19 genomic sites within four gene loci will help the community to assess the strengths and challenges of CRISPR/Cas9 gene editing in the mouse.

Reviewer

This revised manuscript presents more data on experimental protocols and mutant sequences, but I had a very hard time determining what was actually done.

1. For example, nowhere in the text do the authors say that they have used multiple sgRNAs for many of the individual sites shown in Figure 1. This information was only available in the supplementary materials. That made it difficult to understand the difference between “several sgRNAs targeting a single site” and “sgRNAs targeting several sites.” Define a “site.”

Response

We have now included a section in the first paragraph of the Results that clarifies the experimental approaches taken. We also included three figures (Figures 2-4) in the main text that outline our detailed approaches to delete individual sites and multiple sites, using both one-step and two-step approaches. A “site” is defined as a specific genomic sequence that has been targeted for deletion. For example, each of the STAT5 binding sites (GAS motifs) is considered a “site”.

2. Throughout the manuscript, the authors should state whether they are analyzing only independent mutations – i.e., ones derived from separate founders. Was sequencing done directly on founders, or only on F1 or F2 animals, as suggested in the Methods section?

Response

We have analyzed independent mutations derived from separate founders and we have also stated this in the text (page 5, line 1-2). We have sequenced each of the targeted sites in all of the ~630 founders. We have also sequenced the targeted sites in heterozygous mice (F1) and homozygous mutants from >50 lines, i.e. mice from the F2 generation. Some of the founders were mosaic and from several founders we generated two or more lines.

3. The nomenclature for experiments listed in Table S5 is very obscure. What are the “Mutant Lines” into which injections were done? After much effort, I gather that Mutant Line B was injected separately with sgRNAs for site B (cf Fig. 1), but was the recipient already mutant?

Response

We changed the nomenclature ‘Mutant line’ to ‘Targeted genomic sites’ in Table S5 for reasons of clarity. The genomic site B (two adjacent STAT5 binding sites in the *Socs2* promoter) was targeted independently with two sgRNAs (1 and 2) in wild type mice. We have also indicated the exact targeting strategy for each genomic site, which are “one-step” or “two-step” deletions. For example, in the “one-step” approach, sgRNAs for two sites are co-injected into zygotes and in the “two-step” approach we initially targeted one site, generated homozygous mice and used their zygotes as hosts to target the second site. We have now also stated the exact names of the mutant lines that carried deletions in one specific site and were subsequently used as recipients to introduce a mutation into a second site within the same gene locus (Figure 2, 3, and 4). This was the only reliable strategy in mutating juxtaposed STAT5 enhancers within the Wap super-enhancer.

4. Information on Line C-2/3 In Table S2 suggests that embryos already mutant at site C-3 were injected with sgRNA #1 for site C-2. What about D-1/2/3/4? Was it already mutant? For what? And what sgRNAs were injected? Similarly for D-1/3/4, for example. Which sites were targeted, and what was the background? In Figure 4b, are these data for multiple sgRNAs targeting a single site, or multiple sites? How far apart are the sites?

Response

We have now shown in a diagram (Figure 3) the generation of the D-1/3/4 and D-1/2/3/4 mutants. D-1/3/4 was generated by coinjecting sgRNAs targeting sites 1, 3 and 4 and we obtained one line that carried individual deletions in these three sites. We generated homozygous D1/3/4 mice and injected their zygotes with two sgRNAs for site 2 and we obtained D-1/2/3/4 mice.

In our previous Figure 4b (current Figure 7b), we used data for more than one sgRNA targeting individual genomic sites (Figure 3). In case of targeting more than one genomic site with several sgRNAs, we have now indicated the distance between each targeting sites in Figure 4.

5. In my view the authors continue to exaggerate the novelty of their observations. There are many more examples of microhomology-mediated deletions than are shown in Table S1a.

Response

We have now included five more examples of microhomology-mediated deletions in various organisms including *Drosophila*, zebrafish, and some parasites in Table S1a. We agree that microhomology-mediated deletions (deletion with repeat sequences aligned at one end) have been reported in previous studies, but to date the frequency of such deletion patterns had not been examined systematically in a large cohort (122 founder mice) and deletions with repeat sequences aligned at both ends had never been reported.

6. No attempt has been made to compare current and prior results in any detail, perhaps in a scatter plot of deletion sizes. Such a plot for asymmetric deletions would have been welcome in Figure 2, beside that for “symmetric deletions.” Asymmetric deletions have been observed repeatedly as well, and a specific comparison with current experiments is needed.

Response

To address the frequency of asymmetric deletions, we have conducted a thorough literature search and identified 18 papers that describe mice whose genome had been targeted with single sgRNAs (Supplementary table 1). We retrieved all the raw data sets from these papers and analyzed them (Supplementary table 6). Eleven of these did not provide sequence data. Out of the remaining seven only the paper by Kim’s group (PMID 24253447) reported a sufficiently large number of sequences (84) to permit an analysis. We analyzed Kim’s datasets and determined a rate of asymmetric deletions with two different cutoffs, 1.5-fold between the upstream and the downstream of Cas9 cutting site and 2-fold differences between each side. With both cutoffs, asymmetric deletions were prevalently observed in both Kim’s datasets and ours. With a cutoff of 1.5-fold, 89% were asymmetric deletions in 84 founder mice targeting two genomic sites compared to 82% in our study with 139 founder mice targeting nine genomic sites. With a cutoff of 2-fold, 82% were asymmetric deletions in Kim’s datasets compared to 73% in our study. This analysis has been included in the result part and Supplementary Figure 1a.

We have also compared deletions sizes of asymmetric deletions obtained from Kim’s datasets and from ours (Supplementary Figure 1b). Although the median size of asymmetric deletions was similar (Kim’s: 11.5 bp and ours: 10 bp), the maximum deletion size was 585 bp in our study compared to 269 bp in Kim’s and the deletion sizes of top 50% was bigger in our datasets. Although we have also tried to compare the deletion sizes of symmetric deletions between Kim’s datasets and ours, the number of founder mice that show the symmetric deletions was only a few (7 founders) in Kim’s datasets, which is not comparable to ours (28 founder mice).

We used relevant search terms and were unable to identify papers that have described “asymmetric deletions” in the mouse genome. Kim’s group had not analyzed their data but our analysis revealed asymmetric deletions in his experiments.

7. The definition of what is asymmetric continues to be arbitrary. The bias toward 5’-directed deletions has also been reported, and the bias shown here – 59% vs. 41% – is not large enough to have a strong effect on target choice.

Response

In addition to a 1.5-fold difference cutoff we have also calculated a 2-fold cutoff. Notably, using a 2-fold difference as a cutoff, more than 70% of deletions were defined as asymmetric. This is stated on page 5. Although 59% of deletions extend towards the 5’ end, this is not statistically different ($p=0.6$) from the 41% at the 3’ end.

8. The value of making mutations at individual sites with sequential injections to avoid very large deletions seems obvious. Some other information is missing. The text should mention that insertions were found at the cut sites as well, perhaps just giving their frequency and any particularly notable features.

Response

After having seen the actual results of simultaneous versus sequential deletions it is obvious that the latter approach is the only reasonable one to delete adjacent sites. We believe that the results from our studies and previous work make it clear that co-targeting adjacent sites will result in the complete deletion of intervening sequences.

We have now added an entire paragraph in the Result section describing insertions that we have observed and stated their frequency in the three distinct targeting strategies, 1) targeting individual genomic sites with single sgRNAs, 2) targeting individual genomic sites with more than one sgRNA, and 3) targeting more than one genomic site with several sgRNAs (page 8-9). Notably, we have observed two different types of insertions, which are insertion combined with deletions (Type A) and insertion only (Type B) (Supplementary Table 2, 3, and 4, insertion sequences were indicated in red). Although most insertions consisted of only a few nucleotides, we also observed a large insertion of 800 nucleotides that was comprised of repetitive sequences.

9. In the cases of large deletions, don't some of the junctions look like they are microhomology-based?

Response

No, we have not seen microhomology-based large deletions.

10. There are some errors to be corrected – e.g., the continuous large deletions illustrated in Fig. S5 appear to come from D-1/2/3/4/5 (see Table S4), not from E-1/2/3/4/5 as stated. On p. 3, it is not made clear the PAM is outside the sequence complementary to the sgRNA. On p. 6, it says the longest deletion from a single sgRNA was 6 kb, while a maximum of 0.6 kb is given elsewhere.

Response

We apologize for the oversight and have corrected the errors.

11. Overall, this paper provides insufficient information on how experiments were done and which experiments are described in each figure or in the text.

Response

As outlined throughout the response letter we have now done our best to provide all information requested by the reviewer, including clear diagrams outlining the targeting strategies.

We have also decided to change the title of the manuscript to make it more pertinent to the mouse genetics community. We chose "Comprehensive assessment of CRISPR/Cas9 induced deletions at 19 sites in the mouse genome".

Reviewers' Comments:

Reviewer #1:

Remarks to the Author:

Reviewer 3 raises some reasonable objections, although I still believe the body of experiments performed and the information provided could be useful to the scientific community. The authors have done a good job in my view placing their work in the context of previous papers. The issue of novelty is certainly present, but it is an editorial decision in the end.

Several of the criticisms raised by the reviewer could be addressed by the authors in a rapid revision, as they do not require performing additional experiments.

More specifically:

- * Clarification of what they mean by sites (16 sites were tested, not 19, as far as I understand)
- * The method section should be expanded to address the concerns and the requests for clarification from reviewer 3.
- * How many guides were used for each site?
- * Figure 5f: the panel is misleading because it states that there is a preferential deletion 5' to the gRNA, while in the text it is stated that such a difference is not statistically significant. I would remove the panel or, better, replace the title and include the p value for the difference.
- * Did the authors observe any inversion when using multiple sgRNAs simultaneously? These are predicted to occur at a noticeable frequency and unless PCR primers are specifically designed to detect them they will be missed. This should be discussed.

I couldn't verify other criticisms raised by the reviewer. In particular, reviewer 3 says that the authors claim on page 6 that the largest deletion obtained with single gRNA is 6kb, but I cannot find this statement on page 6 or elsewhere in the manuscript. It seems to me that they consistently state that the largest deletion is ~ 0.6 kb .

Also, the reference to a mistake on a large continuous deletions illustrated in Figure S5 is unclear to me. The figure shows a scatter plot of deletion size by GC content, not a specific deletion. Perhaps he refers to figure S6, but even in this case I don't know how he can conclude that the deletion is refers to D-1/2/3/4/5 and not to E-1/2/3/4/5 (Supplementary Table 4 was a bit confusing for me to figure this out!). I would be curious to know how the authors respond to these concerns.

Response to reviewers

Reviewer

1. Clarification of what they mean by sites (16 sites were tested, not 19, as far as I understand)

Response

Including our latest experiment we have now targeted a total of 17 genomic sites. We defined a “target site” as a stretch of DNA (~100 bp) recognized in ChIP-seq experiments by specific transcription factors or by CTCF. In case of transcription factor binding sites (mainly STAT5) we targeted the proposed DNA recognition motifs. For CTCF binding sites we targeted the area covered by CTCF in ChIP-seq experiments, generally between 100 and 200 bp.

Including our most recent data, we targeted six STAT5 binding sites in three genomic loci (*Socs2*, *Stat5*, and *Wap*), ten CTCF binding sites in two genomic loci (*Wap* and *Csn*), and one NFIB binding site in one genomic locus (*Csn*), individually and in relevant combinations as shown in Figure 1. Thus, we targeted a total of 17 individual sites in four gene loci. Stepwise deletions were used to ablate more than one specific site in a given locus, which led to a total of 19 targeting experiments.

2. The method section should be expanded to address the concerns and the requests for clarification from reviewer 3.

Response

We have now described the targeting strategies in more detail. We have included sgRNA numbers in the Method section under ‘Design of sgRNA/Cas9 and microinjection into mouse zygotes’.

3. How many guides were used for each site?

Response

The number of guides and the sequences of the guides for each target site had already been provided in supplemental Tables 2, 3 and 4. We have now included the number of guides used for each site in the Methods section (page 13, highlighted in yellow).

4. Figure 5f: the panel is misleading because it states that there is a preferential deletion 5' to the gRNA, while in the text it is stated that such a difference is not statistically significant. I would remove the panel or, better, replace the title and include the p value for the difference.

Response

We apologize for the confusing statement. We changed the wording in the figure panel from ‘preferential deletion 5' to the gRNA’ to ‘proportion of deletion to 5' and 3'-end of sgRNA’.

5. Did the authors observe any inversion when using multiple sgRNAs simultaneously? These are predicted to occur at a noticeable frequency and unless PCR primers are specifically designed to detect them they will be missed. This should be discussed.

Response

We did not design primers to detect potential inversions. Having said this, in the experiment where we deleted a >20 kb region in the *Wap* locus (targeted genomic site: D-1/2/3/4/5) all founders appeared to carry homozygous or compound heterozygous deletions. Thus, at least in this locus and in this experiment, we did not obtain inversions. We have now addressed the possibility of inversions in the discussion (page 11, highlighted in yellow).

6. I couldn't verify other criticisms raised by the reviewer. In particular, reviewer 3 says that the authors claim on page 6 that the largest deletion obtained with single gRNA is 6kb, but I cannot find this statement on page 6 or elsewhere in the manuscript. It seems to me that they consistently state that the largest deletion is ~ 0.6 kb.

Response

We had corrected this error in the early revision (page 7).

7. Also, the reference to a mistake on a large continuous deletions illustrated in Figure S5 is unclear to me. The figure shows a scatter plot of deletion size by GC content, not a specific deletion. Perhaps he refers to figure S6, but even in this case I don't know how he can conclude that the deletion is refers to D-1/2/3/4/5 and not to E-1/2/3/4/5 (Supplementary Table 4 was a bit confusing for me to figure this out!). I would be curious to know how the authors respond to these concerns.

Response

Figure S5 mentioned by reviewer #3 is Figure S6 in the current version and we corrected the typo E-1/2/3/4/5 to D-1/2/3/4/5 in the previous revision. Following the reviewers' comments, we provided detailed information in Supplemental Table 4. We included the diagrams how we deleted multiple genomic sites in stepwise for the clarity.